# The Effects of Vitamin D on the Expression of IL-33 and Its Receptor ST2 in Skin Cells; Potential Implication for Psoriasis

**DOI:** 10.3390/ijms222312907

**Published:** 2021-11-29

**Authors:** Justyna M. Wierzbicka, Anna Piotrowska, Dorota Purzycka-Bohdan, Anna Olszewska, Joanna I. Nowak, Aneta Szczerkowska-Dobosz, Bogusław Nedoszytko, Roman J. Nowicki, Michał A. Żmijewski

**Affiliations:** 1Histology Department, Faculty of Medicine, Medical University of Gdansk, 80-210 Gdansk, Poland; justyna.wierzbicka@gumed.edu.pl (J.M.W.); annapiotrowska@gumed.edu.pl (A.P.); anna.olszewska@gumed.edu.pl (A.O.); j.chorzepa@gumed.edu.pl (J.I.N.); 2Department of Dermatology, Venereology and Allergology, Faculty of Medicine, Medical University of Gdansk, 80-214 Gdansk, Poland; dorota.purzycka-bohdan@gumed.edu.pl (D.P.-B.); aneta.szczerkowska-dobosz@gumed.edu.pl (A.S.-D.); boguslaw.nedoszytko@gumed.edu.pl (B.N.); roman.nowicki@gumed.edu.pl (R.J.N.); 3Invicta Fertility and Reproductive Centre, Molecular Laboratory, 80-850 Gdansk, Poland

**Keywords:** vitamin D, IL-33, ST2, skin, psoriasis

## Abstract

Interleukin 33 (IL-33) belongs to the IL-1 family and is produced constitutively by epithelial and endothelial cells of various organs, such as the skin. It takes part in the maintenance of tissue homeostasis, repair, and immune response, including activation of Th2 lymphocytes. Its involvement in pathogenesis of several inflammatory diseases including psoriasis was also suggested, but this is not fully understood. The aim of the study was to investigate expression of IL-33 and its receptor, ST2, in psoriasis, and the effects of the active form of vitamin D (1,25(OH)_2_D_3_) on their expression in skin cells. Here we examined mRNA and protein profiles of IL-33 and ST2 in 18 psoriatic patients and healthy volunteers by qPCR and immunostaining techniques. Potential effects of 1,25(OH)_2_D_3_ and its receptor (VDR) on the expression of IL-33 and ST2 were tested in cultured keratinocytes, melanocytes, fibroblasts, and basal cell carcinoma cells. It was shown that 1,25(OH)_2_D_3_ effectively stimulated expression of IL-33 and its receptor ST2’s mRNAs in a time-dependent manner, in keratinocytes and to the lesser extends in melanocytes, but not in fibroblasts. Furthermore, the effect of vitamin D on expression of IL-33 and ST2 was VDR-dependent. Finally, we demonstrated that the expression of mRNA for IL-33 was mainly elevated in the psoriatic skin but not in its margin. Interestingly, ST2 mRNA was downregulated in psoriatic lesion compared to both marginal tissue as well as healthy skin. Our data indicated that vitamin D can modulate IL-33 signaling, opening up new perspectives for our understanding of the mechanism of vitamin D action in psoriasis therapy.

## 1. Introduction

Beyond the effects on bone homeostasis, vitamin D exerts an impact on a variety of other organs or systems, including the skin and immune system. Keratinocytes are not only a natural source of vitamin D, but due to the expression of the vitamin D receptor (VDR), they are also target cells, which respond in an autocrine and paracrine manner [1]. They are the only cells that can produce vitamin D from its precursor 7-DHC [2], and which are equipped with all the enzymes (CYP2R1, CYP27A1 and CYP27B1) necessary to metabolize vitamin D into its active metabolite— 1,25-dihydroxyvitamin D_3_ (1,25(OH)_2_D_3_) [3,4].

Vitamin D exerts a number of actions in the skin, for example: it regulates cell proliferation, differentiation, apoptosis, and modulates immune activity [5,6]. Most of these actions are mediated by the vitamin D receptor (VDR), which, after 1,25(OH)_2_D_3_ binding, interacts with the retinoid X receptor (RXR) and promotes the expression of the target genes [7]. A nongenomic pathway activated by vitamin D, with the potential involvement of an alternative vitamin D binding protein, PDIA3, has been also described [8]. Along with advances in molecular and clinical studies, vitamin D has gained attention as a potential player in the pathogenesis of various skin diseases, including psoriasis.

Psoriasis is a chronic skin disorder characterized by scaly, red lesions resulting from abnormal keratinocytes proliferation and cutaneous inflammation, accompanied by skin vessels dilation [9,10]. Skin psoriatic lesions are the most common manifestation of psoriasis, although inflammatory processes can also develop in other locations, such as intestines or the cardiovascular system [11]. A critical role for vitamin D in psoriasis is underlined by many studies, which report either a vitamin D deficiency or insufficiency in psoriatic patients [12,13]. However, considering the high prevalence of vitamin D deficiency worldwide, some studies show no significant difference between psoriatic patients and healthy controls [14]. Overall, the evidence point to an inverse correlation between serum concentration of 25(OH)D and the severity of the disease in psoriatic patients, which may significantly contribute to the pathogenesis of this dermatosis [15,16,17,18].

As a consequence of the the anti-proliferative and anti-inflammatory actions of vitamin D [6,19,20], it can reverse the abnormal epidermal function related to this severe skin condition. Psoriasis is a remarkable example of a non-skeletal disease in which active forms of vitamin D as well as its analogues have proved to be effective therapeutics.

Psoriasis most commonly manifests on the skin, although inflammatory processes can occur also in other organs [21,22]. Indeed, psoriasis is currently considered a systemic pathology, which can include other conditions, such as psoriatic arthritis to obesity and metabolic disease (MetS), with increase cardiovascular risk [23,24,25]. Obesity seems to be a key psoriasis comorbidity, leading to a higher prevalence of cardiovascular disease. Epidemiological studies reported a variable frequency of obesity ranging from 15 to 30% among psoriasis patients, however, most studies only included individuals of Caucasian ancestry [26]. Lipid abnormalities are significantly prevalent among psoriatic patients. Severe psoriasis is associated with approximately 1.36–5.55 times greater risk of dyslipidemia [27]. Large-scale meta-analysis of population-based studies carried on the study group of 40,000 psoriatic patients demonstrated a significant link with metabolic syndrome with a more than 2-fold susceptibility risk in comparison to the general population [28]. Moreover, the prevalence of comorbidity was psoriasis severity-dependent [28]. Mood disorders and major deterioration in the quality of life are reported to be a common issue among psoriatic patients (up to 62% prevalence) [27]. Other factor, such as sleep disturbance, caused by jetlag, may influence disease severity and DLQI (Dermatology Life Quality Index) scores [29]. The mood disorders may have their originate from either the direct pro-inflammatory background or the psoriasis burden. Interestingly, the cytokine pro-inflammatory theory of depression, suicidality and psoriasis overlap [30].

Histologically, the dermatosis is characterized by hyperproliferation of keratinocytes, impaired epidermal barrier function at the sites of skin lesions, and skin infiltration by activated inflammatory cells [25,31]. The etiology of psoriasis is not fully understood. Several factors contribute to its development, such as auto-immunological, genetic, hormonal and psychosomatic issues [23,25].

The recently discovered IL-33/ST2 signaling pathway has been linked to a wide variety of inflammatory skin disorders, including psoriasis and atopic dermatitis [32,33,34]. Paradoxically, while psoriasis is considered as a Th1 and Th17 immunological disorder [35], there are some reports suggesting the pro-inflammatory contribution of IL-33 to this skin disease [36,37,38].

IL-33, originally identified as NF-HEV, a nuclear factor from high endothelial venules [32], is a new member of the IL-1 family [39]. Unlike other family members, IL-33 is typically expressed in epithelial cells (keratinocytes in epidermis and pulmonary epithelial cells, etc.) and fibroblasts exposed to environmental stimulants, rather than in hematopoietic cells [32,39]. Interestingly, the high level of constitutive expression of IL-33 is characteristic for human tissues under physiological conditions [40]. This cytokine is typically found in the nucleus, serving as an transcription factor [32,37,41], however, it may also act as an alarmin—cytokine released from epithelial cells after tissue injury [32]. Nuclear IL-33 represses gene expression by promoting chromatin condensation [42]. Furthermore, it is postulated that nuclear IL-33 retention serves as a fundamental regulator of its extracellular activity, preventing its uncontrolled release outside the cell [39]. Indeed, in chimeric mice the latter was shown to induce a non-resolving lethal inflammation [43]. 

IL-33 has two forms; a full-length IL-33 (proIL-33) and a mature IL-33, processed by caspases [39,44]. The full-length IL-33 has a nuclear localization, containing an N-terminal nuclear localization sequence as well as a chromatin-binding motif [33,44]. However, this proIL-33 can also be released into the extracellular space upon tissue damage, cellular stress or, eventually, cell death [40]. Extracellularly, the full-length IL-33 serves as an alarmin cytokine, alerting various types of immune system cells, especially type 2 innate lymphoid cells (ILC2, natural helper cells) [40], inducing a Th2-type inflammatory response [37]. Surprisingly, in contrast to other IL-1 family members, cleavage of IL-33 within its C terminal domain by caspases does not activate it, but rather results in an inactivation which occurs during apoptosis [45]. Nevertheless, the full-length IL-33 may not be the only active form, because it may also be cleaved by enzymes produced by a variety of inflammatory cells [32]. For instance, cleavage of IL-33 by enzymes released from neutrophils, such as elastase, cathepsin G or proteinase 3, results in the formation of new 18 kDa and a 21 kDa bioactive mature IL-33 forms, characterized by a high potency to IL-1 stimulation of the family orphan receptor [46,47]. Therefore it has been suggested, that the inflammatory environment within the damaged tissue influences IL-33 bioactivity [32]. The cytokine function of IL-33 relies on binding to the heterodimeric membrane-bound orphan IL-1 family receptor ST2 from TLR/IL1R superfamily [48]. Different isoforms of the ST2 receptor are present in a large variety of innate and adaptive immune cells, including eosinophils, basophils, type 2 T-helper cells, dendritic cells, NK cells, mast cells and, keratinocytes [32,49,50,51].

Given the recognized involvement of IL-33 in psoriasis, the immunosuppressive activity of vitamin D and its therapeutic role in psoriasis, in the present study we investigated the effects of vitamin D on the expression of IL-33 and its receptor (ST2).

## 2. Results

Vitamin D is a known regulator of immune response. Thus, we investigated the effects of the major active form of vitamin D (1,25(OH)_2_D_3_) on the expression of IL-33 at mRNA and protein levels (Figure 1 and Figure 2) in skin-derived cell lines: keratinocytes, melanocytes, fibroblasts, and basal cell carcinoma cells. 1,25(OH)_2_D_3_ effectively stimulated the expression of *IL-33* and its receptor *ST2* mRNAs in a time-dependent manner in keratinocytes and to a lesser extends in melanocytes (Figure 1A,B). Fibroblasts did not express either in *IL-33* or *ST2* genes. In the A431 basal cell carcinoma cell line, the expression of *IL-33* and *ST2* was significantly stimulated by 1,25(OH)_2_D_3_ at all time points with the greatest effect after 4 h for *IL-33* and after 8h for *ST2*. Since cytokines, such as TNF, IL-1B, IFNG, and IL-17A, stimulate the production of IL-33, we also examined the effect of 1,25(OH)_2_D_3_ on their expression (Figure 1C–F). All of the tested genes examined were expressed in keratinocytes and melanocytes. Expresssion was not observed in fibroblasts, with the only exception being *IL-1B* mRNA, where a low level of mRNA was detected but not altered by 1,25(OH)_2_D_3_. In primary keratinocytes, 1,25(OH)_2_D_3_ moderately but significantly induced the expression of *TNF* at all time points, while the expression of *IL-1B* was induced only after 4 h of treatment. The expression of *IFNG* was not affected by 1,25(OH)_2_D_3_, while *IL-17A* was downregulated after treatment for 24 h. In melanocytes, 1,25(OH)_2_D_3_ treatment resulted in significantly upregulation of the expression of *IL-1B*, *IFNG* and *IL-17A* after 4 and 8 h treatment and *TNF* after 24 h. Basal cell carcinoma cell line did not express the *IL-17A* gene, while the expression of the other genes examined was significantly upregulated only after 4 h of treatment with 1,25(OH)_2_D_3_.

IL-33 protein levels were analyzed by Western blotting and immunofluorescence staining (Figure 2). Western blot analysis revealed differences in the apparent molecular mass of the processed IL-33 protein, implying the presence of post translational modifications (Figure 2A). IL-33 is primarily produced as a 31 kDa precursor (pro-form), which is digested by proteases, such as elastase, cathepsin G, tryptase or chymase into ~18–20 kDa mature forms [47,52]. Interestingly, a substantial amounts of full-length IL-33 (~31 kDa) was only seen in the primary keratinocytes with an additional band (~22 kDa) was appearing after 1,25(OH)_2_D_3_ treatment. The processed, mature form of IL-33 protein (~19 kDa) was detected in all cell lines, with the lowest amount in fibroblasts. In melanocytes and fibroblast, 1,25(OH)_2_D_3_ has not affected on the IL-33 level, and in keratinocytes, there was a moderately increased the protein level.

Intracellular localization for IL-33 was mainly limited to the nuclei of control keratinocytes as shown by immunofluorescent staining (IF). The 1,25(OH)_2_D_3_ treatment did not induce the translocation of IL-33 to the cytoplasm, and the majority of IL-33 immunoreactivity remaining in nuclei (Figure 2B). The staining for ST2 shows cytoplasmatic localization of protein, and 1,25(OH)_2_D_3_ treatment did not affect its localization, but significantly increased its level, especially in differentiating keratinocytes.

Since our studies showed that 1,25(OH)_2_D_3_ can greatly stimulate expression of IL-33 and ST2 in keratinocytes, we further tested whether its effects is mediated by the receptor for vitamin D (VDR). In order to discriminate the VDR-dependent genomic pathway from alternative fast response pathways to vitamin D, the HPEKp cell line was transfected with either a specific VDR-siRNA or a non-specific siRNA (Figure 3). Thus, the transfection allowed us to assess the effect of 1,25(OH)_2_D_3_ on cells with reduced VDR expression. The degree of VDR knockdown was assessed at the mRNA level with a substantial reduction of *VDR* transcripts being observed (Figure 3A). Since CYP24A1 is strongly upregulated by the complex of VDR and 1,25(OH)_2_D_3_, we used real-time PCR to confirm if *CYP24A1* gene transcript was downregulated. In control cells, 100 nM 1,25(OH)_2_D_3_ strongly induced *CYP24A1* expression, whereas in VDR-siRNA transfected cells, this induction of *CYP24A1* expression was strongly decreased (Figure 3B). Both results were confirmed at the protein level by immunoblotting (Figure 3C). Next, we found that the expression of IL-33 and ST2 was downregulated at both mRNA (Figure 3D,E) and protein levels (Figure 3F) in the HPEKp cell line with silenced VDR expression, suggesting that VDR is essential for 1,25(OH)_2_D_3_ dependent modulation of their expression. To confirm these results, we used A431 cell line with VDR knockdown (A431 ΔVDR) generated by CRISPR/Cas9 technology (Figure 4). Although limited expression of *VDR* (Figure 4A) was found, due to the introduction of a frame shift mutation, we did not observe the protein product of the *VDR* gene (Figure 4C). Furthermore, after treatment of A431 cells with VDR knockdown with 1,25(OH)_2_D_3_, no CYP24A1 transcript or protein was observed (Figure 4B,C). Analogously to the HPEKp cell line, in A431 cell line an effect of VDR knockout on the expression of the genes under study was observed. Both mRNA (Figure 4D,E) and protein (Figure 4F) expression of IL-33 and ST2 were significantly downregulated after 1,25(OH)_2_D_3_ treatment in A431 ΔVDR, compared to the wild type A431 cells.

Our data indicated that vitamin D stimulation of theexpression of IL-33 and ST2 requires functional VDR receptor, thus we checked for potential VDR binding sites in the promotor sequences of genes under study. For this purpose, we used the CiiiDER program [53] and all position frequency matrices acquired from JASPAR database [54]. The results of the analysis of the DNA sequence upstream of *IL-33* coding region, showed the presence of one VDR binding site and a matrix score of 1 (chromosome 9: 24.635–24.642). Moreover, 44 VDR binding sites, which had a matrix score from 0.98 to 0.85, were found in the first 10 kb upstream of ATG start site, and a further 44 VDR binding sites 10–20 kb upstream of ATG, which also a matrix score from 0.98 to 0.85 (Figure 5A). All potential VDR sites were of the MA0693.2 JASPAR type. In the case of the *ST2* promotor, we found two VDR binding sites in the first 10 kb upsteam of the start site with a matrix match score = 1 (chromosome 2: 24,635–24,642; 20,484–20,491) and 47 VDR binding sites with a matrix match score from 0.98 to 0.85. For the sequence 10 kb upstream from the start site we also found two binding sites that had a score of 1 (chromosome2: 18,643–18,650; 15,457–15,450) and 36 VDR binding sites with a score from 0.98 to 0.85. Additionally, in the *ST2* sequence, we found the RXRBvar2 (MA1556.1) sequence with the matrix match score 0.85 (chromoseome2: 18,657–18,644) and 0.87 (chromosome2: 9365–9378) (Figure 5B). All potential VDR binding sites generated with the CiiiDER program are included in the Appendix A. Finally, our predictions were confirmed by in silico analysis of available ChiP-Seq data (Appendix A).

Skin biopsies of psoriatic lesions and potentially unaffected margin area were taken from 18 volunteers from the psoriasis group and from 12 individuals from a control group with healty skin (Figure 6). It was shown that the expression of mRNA for *IL-33* was mainly elevated in psoriatic skin but not in its margin (Figure 6A). Interestingly, the opposite effect was observed by immunodetection of specific IL-33 epitopes in skin biopsies (Figure 6C,D). A statistically significant increase in IL-33 immunoreactivity, but not in mRNA level, was observed in the margin of psoriatic lesions but not in the lesional skin. Interestingly, it seems that the IL-33 immunoreactivity is preferentially localized in the nucleus or perinuclear area in the affected skin, while more cytoplasmic IL-33 immunoreactivity was observed in the marginal tissue and skin of healthy controls. Finally, strong perivascular immunoreactivity characteristic for IL-33 was detected in lesional skin. It seems that mononuclear cells infiltrating the perivascular areas of the dermis also produce IL-33 and this can contribute to the overall elevation in the level of *IL-33* mRNA detected in psoriatic skin, and is positively correlates with an increased levels of protein. Moreover, the mRNA level of *ST2* was measured in the above skin biopsies. In contrast to the upregulated expression of *IL-33*, *ST2* was downregulated in psoriatic lesion compared to both marginal tissue and healthy skin (Figure 6B). Immunofluorescence staining for the ST2 receptor confirmed RT-PCR results. Interestingly, the highest immunoreactivity was observed in stratum granulosum and corneum; within the deeper layers of the epidermis, where staining intensity was moderate. The *IL-33* and *ST2* mRNA levels in the margin of psoriatic lesions as well as within plaques did not correlate with the PASI, BSA or DLQI score (data not shown).

## 3. Discussion

IL-33 belongs to the IL-1 family, and it is constitutively expressed in epithelial barrier tissues and lymphoid organs, where it plays an important role in type-2 innate immunity through its receptor ST2. IL-33 appears to function as an alarmin (alarm signal) rapidly released upon cellular damage or inflammation [40]. IL-33 can act as both a pro- and anti-inflammatory factor [55]. Under homeostasis conditions, IL-33 maintain tissue integrity, limiting excess inflammation, and promoting tissue adaptation to remodeling and other stressors. It has been reported that full-length IL-33 interacts with the transcription factor NF-κBforming a complex which reduces the delay in the expression of NF-κB target genes, such as *IκBα* and *TNF*, and therefore diminishes pro-inflammatory signaling [56]. Interestingly, 1,25(OH)_2_D_3_, a known factor contributing to the switch between the Th1 to Th2 immune response, stimulates the expression of *IL-33* mRNA, as well as its receptor, *ST2,* in human primary keratinocytes. This provides some mechanistic explanation as to how vitamin D stimulates the Th2 response. Furthermore, the classic inducer of keratinocytes differentiation, calcium, did not stimulate the expression of *IL-33* mRNA (data not shown). It has been demonstrated that some cytokines can modulate the release of IL-33. In a study using ex vivo psoriatic skin organ cultures, it was observed that stimulation with TNF increases *IL-33* mRNA expression in comparison to untreated psoriatic skin [57]. Other studies performed on normal human epidermal keratinocytes (NHEK) demonstrated that not only TNF but also INFG induces the expression of *IL-33* [58]. In further studies, the same authors showed that IL-17A strongly upregulates the expression of *IL-33* in the NHEK cells, suggesting involvement of IL-33 in the pathophysiology of psoriasis [49]. One of the newest studies, one showed that intradermal injection of recombinant IL-33 alone can induce psoriasis-like dermatitis through the transcriptional upregulation of such genes as *IL-17*, *TNF*, and other pro-inflammatory chemokines [38]. However, in our studies, *IL-17A* was downregulated in HPEKp cells, therefore we decided to investigate whether there is a direct role for the vitamin D receptor in *IL-33* and *ST2* gene expression. Furthermore, VDR knockdown in HPEKp cells and knockout in the A431 cell line blocked the stimulatory effect of 1,25(OH)_2_D_3_, confirming that this effect is mediated by the VDR. Finally, the presence of potential VDR binding sites in the promotor region of the genes under study were confirmed *in silico*. Therefore, it can be postulated that the effect of 1,25(OH)_2_D_3_ in altering *IL-33* and *ST2* mRNAs expression is mediated by VDR and is not connected with induction of keratinocytes differentiation. Immunofluorescent staining did not reveal the translocation of IL-33 from the nucleus to the cytoplasm after 1,25(OH)_2_D_3_ treatment, suggesting that vitamin D primarily regulates IL-33 expression at mRNA and protein levels.

Psoriasis is a common, long-lasting autoimmune skin disorder characterized by patches of abnormal proliferating keratinocytes covered by a silvery, white scale. So far, there is no cure for psoriasis; however, numerous treatments can alleviate the symptoms. The first line of treatment is topical corticosteroids. Currently, topical vitamin D as well as it analogs are also widely used in psoriasis treatment, either as monotherapy or in combination with topical steroids due to their synergistic, complementary effects [59].

A major question addressed in this paper is how vitamin D-mediated stimulation of the expression of IL-33 and ST2, contributes to the treatment of psoriasis, where as we have presented the expression of the above is highly dysregulated. As widely known 1,25(OH)_2_D_3_ decreases the rate of proliferation of keratinocytes and regulates their differentiation. Furthermore, vitamin D inhibits the secretion of pro-inflammatory cytokines, such as IL-2, IL-6, and IL-8, and by doing so, it reduces T-cell proliferation and induces regulatory T-cell differentiation [60]. In psoriasis, an imbalance in the T-cell immunophenotype was observed and the ratio of Th1/Th17 cells was found to be increased, and Th2/T-reg cells were downregulated, compared with healthy individuals [61]. T-reg cells are one of the primary tissue-resident cells that constitutively express high levels of ST2 and this makes these cells a suitable target for IL-33 [55]. Interestingly, in psoriatic plaques but not in margin tissue or control healthy skin, perivascular mononuclear cells preferentially express the cytoplasmic form of IL-33, which, according to the literature, is secreted and may contribute to the development or aggravation of the symptoms of psoriasis.

It is well established that activation of the Th1 type of immune response is the hallmark of psoriasis, thus it seems that activation of a pro Th2 response by IL-33 should attenuate psoriasis. However, our results indicated that there is a significantly higher *IL-33* mRNA levels in psoriatic lesions and encoded protein in its potentially disease-free regions (margin tissue). IL-33 was present in both the nucleus and cytoplasm of psoriatic keratinocytes. Our results are in agreement with others research groups, which observed elevated levels of *IL-33* mRNA in affected psoriatic skin, compared with healthy skin [36,37,38]. The protein level of IL-33 was marginally decreased in psoriatic lesions. That observation may be explained by the fact that upon psoriasis inflammatory stimuli, keratinocytes actively release IL-33, which in turn acts on surrounding keratinocytes via an autocrine manner. Autocrine stimulation of keratinocytes results in the production of psoriasis-related cytokines and other inflammatory molecules [38]. It could be speculated that the accumulation of IL-33 in the skin surrounding a lesion could be involved somehow in the progression of the plaque or as a protecting mechanism. However, further studies using the bigger group is needed to evaluate this hypothesis. To respond to IL-33, cells must express the ST2 receptor. Our studies show for the first time that there is a decrease in *ST2* mRNA transcript in psoriasis-affected skin compared with healthy skin. These observations may at least in part, explain the attenuated anti-inflammatory action of IL-33 within psoriatic plaques. Importantly, the decreased expression of *ST2* among psoriatic patients may play a major role in the inhibition of the type 2 inflammatory processes [62]. Our studies show that 1,25(OH)_2_D_3_ is a very strong inducer of *ST2* expression not only in keratinocytes, but also to the lesser extend in melanocytes. A similar effect of vitamin D was observed in cells relevant to asthma by Pfeffer et al. [63]. Our results seem to be contradictory because of the involvement of IL-33 in the development of psoriasis, though the induction of lymphocytes Th17 and IL17 secretion, was recently postulated [38]; however, in our study, expression of *IL-17* was decreased in primary human keratinocytes (HPEKp). In addition, it is well established that vitamin D is a strong inhibitor of Th17 lymphocytes activity [64,65]. Furthermore, the anti-psoriatic properties of vitamin D and its analogs, such as calcipotriol has been shown in imiquimod-induced psoriasis-like dermatitis and the effect was mainly mediated thought the inhibition of T17 lymphocytes expansion [66]. Finally, IL-17 is also involved in the development of pruritus [67], which is common amongst patients with psoriasis [68], and vitamin D and its analogs were found to be effective in the eradication of psoriatic symptoms, including itching [69]. Interestingly, in combination with our data, it could be suggested that the expression of IL-17 could be direct and independent of the IL-33 and ST2 signaling regulation by vitamin D.

Summarizing, the biology of IL-33 is very complex as it can act as both a pro- and anti-inflammatory factor. Our results indicate that there is significant dysregulation of IL-33 and ST2 expression in psoriasis, which may contribute to its pathogenesis. There are a few reports describing an increased secretion of IL-33 by psoriatic keratinocytes and its pro-inflammatory role in psoriasis [32,37,38,70,71]. However, our current work also suggested that there is a contribution from perivascular mononuclear cells to the secretion of IL-33. Our findings also suggest that ST2, a receptor for IL-33, may be involved in psoriasis biology and is a potential target for vitamin D action. Thus, this study may open new perspectives for understanding the mechanisms of vitamin D action in psoriasis therapy.

## 4. Materials and Methods

### 4.1. Cell Cultures

Pooled juvenile Human Epidermal Keratinocyte Progenitors (HPEKp) and Single donor Adult Epidermal Melanocytes (HEMas) were acquired from CELLnTEC (Bern, Switzerland). Cells were cultivated in Epidermal Keratinocyte Medium (CnT-07, CELLnTEC) containing low calcium (0.07 mM), supplement mix (A, B, C), bisphenol A (BPE) and Melanocyte Medium (CnT-40, CELLnTEC), respectively. Both mediums were additionally supplemented with gentamycin and amphotericin. Adult Human Dermal Fibroblasts (HDF) were obtained from Thermofisher Scientific (Waltham, MA, USA), and A431 cell line (basal cell carcinoma) was obtained from Synthego Corporation (Menlo Park, CA, USA); both cell lines were grown in Dulbecco’s Modified Eagle’s Medium (Sigma Aldrich, St. Louis, MO, USA), a high glucose medium (4.5 g/L) supplemented with 10% fetal bovine serum, penicillin, and streptomycin. Cells were cultured at 37 °C in a humidified 5% CO_2_ incubator in T-75 culture flasks. Cells were passaged at 80–90% confluency after trypsinization with TrypLE™ Express solution (Gibco, Thermofisher Scientific, Waltham, MA, USA).

### 4.2. Treatments

In order to study the potentially different responses to 1,25(OH)_2_D_3_ in major cell types of full thickness skin (keratinocytes, melanocytes, and fibroblasts, as well as basal cell carcinoma cells) HPEKp, HEMas, HDF and A431 cells were treated with 0.1 µM 1,25(OH)_2_D_3_ for 4, 8 or 24 h. After the indicated times, cells were subjected to further analysis—real-time PCR, immunofluorescence staining, and/or Western blot. The 1,25(OH)_2_D_3_ was purchased from IsoSciences (Ambler, PA, USA).

### 4.3. Patients and Controls

AEighteen unrelated patients with chronic plaque psoriasis admitted to the dermatology department and to the dermatology outpatient clinic were enrolled. Patients were aged ≥18 years and had not received systemic treatment for psoriasis (cyclosporine, methotrexate, retinoids or phototherapy) for the previous three months or topical anti-psoriatic therapy for the previous one week. Current or previous biological therapies constituted exclusion criteria. Individuals with other chronic skin disorders, psoriatic arthritis, patients on current immunosuppressive therapy, organ transplant recipients or individuals suffering from any other systemic inflammatory diseases or malignancy were excluded from the study participation. Two 4 mm full thickness punch biopsies were collected from every patient, first was from lesional skin and the second from a potentially unaffected border of the lesion. The control group consisted of 13 adult, healthy, unrelated volunteers without psoriasis (also in family history) and free from other chronic inflammatory skin and systemic diseases. In all patients’,a dermatological examination with a psoriasis severity assessment using Psoriasis Area and Severity Index (PASI), Body Surface Area (BSA) and Dermatology Quality of Life Index (DLQI) was performed by the same dermatologist. All participants gave written consent to take part in the study and the study was approved by the local bioethical committee.

### 4.4. RNA Extraction and Real-Time PCR

RNA from cell cultures was extracted using an ExtractMe Total RNA kit (BLIRT S.A., Gdansk, Poland) according to the manufacturer’s instructions. Frozen skin samples (psoriatic skin and matching marginal biopsies, and healthy control samples) were sliced on a microtome to 10 µm thick sections and followed by mechanical homogenization of tissue samples. Total RNA was isolated by the Total RNA Prep Plus kit (A&A Biotechnology, Gdansk, Poland). The concentration and purity of the isolated RNA were measured with an Epoch spectrophotometer (BioTek, Winooski, VT, USA). A total of two micrograms of total RNA were reverse transcribed into cDNA with RevertAid™ First Strand cDNA Synthesis kit (Thermo Scientific, Thermofisher Scientific, Waltham, MA, USA). Real-time PCR reactions were performed in duplicates with SensiFast Sybr™ No-ROX kit (BioLine, London, UK). The data were collected on the StepOnePlus™ Real-Time PCR System (Applied Biosystems, Thermofisher Scientific, Waltham, MA, USA) as previously described [72]. Primers used for PCR amplification are listed in Table 1. The amount of amplified product for each gene was compared to that of the reference gene (*RPL37*) using a comparative ΔΔCT method and presented as a fold change ±SD.

### 4.5. VDR Silencing in the HPEKp Cell Line Based on siRNA Transfection

Knockdown of VDR expression was performed on HPEKp cell line using the Silencer^®^ Select Validated siRNA ID s14777 (Ambion Ltd., Thermofisher Scientific, Waltham, MA, USA). HPEKp cells were transfected with 25 pmol of the indicated siRNAs for 4–6 h using Lipofectamine RNAiMAX (Invitrogen, Thermofisher Scientific, Waltham, MA, USA) according to the manufacturer’s protocol. A negative siRNA (Ambion, cat# 4390844) that had no significant homology to any known gene sequence was used as a control. Cells were assayed 48 h and 72 h after transfection. After this period, the efficacy of VDR knockdown was assessed on mRNA and protein level by real-time PCR and Western bloting.

### 4.6. VDR Gene Knock-Out in the A431 Cell Line Based on CRISPR/Cas9 Technology

To generate VDR knock-out cells, we used the CRISPR/Cas9 system from Synthego Corporation (Menlo Park, CA, USA). Guide RNA was designed for all the VDR splice variants and electroporated in ribonucleoprotein complex Cas9/guideRNA into the cells. Knockdown cells were selected with the sterile cloning disks from the SP Bel-Art (Wayne, NJ, USA). The VDR knockout was confirmed at the mRNA and protein level by real-time PCR and Western bloting.

### 4.7. Immunofluorescence Staining

For immunofluorescence analyses, cells were seeded in 8-well Lab-Tek II chamber slides (Nalge Nunc, Rochester, NY, USA) and after treatments, cultures were fixed with 4% paraformaldehyde (PFA) for 10 min. Frozen skin samples (psoriatic skin and matching marginal biopsies, and healthy control samples) were sliced on a microtome to 10 µm thick sections followed by 10 min fixation in 4% PFA. Further steps were common for cell and skin samples. Specimens were permeabilized for 5 min in 0.2% Triton X100 in PBS. Following washing (3 × 5 min in PBS), slides were incubated for 30 min at RT with 1% BSA in PBS. After blocking primary antibody (rabbit polyclonal anti-IL-33 Bioss bs-2208R and rabbit polyclonal anti-ST2 Sigma-Aldrich PRS3363) diluted in 1%BSA/PBS were applied and specimens were incubated in a humidified chamber at 4 °C overnight. After indicated time, the primary antibody solution was removed, slides were washed three times in PBS and incubated with secondary antibody Alexa Fluor^®^ 488 conjugate (goat anti-rabbit IgG ThermoFisher Scientific A11008) solution in 1% BSA/PBS for 1 h at RT in the dark. Subsequently, slides were washed (3 × 5 min in PBS) and specimens counterstained with DAPI. As negative controls, the primary antibodies were omitted in the procedure. Images were collected with a fluorescence microscope (Nikon Eclipse E800 or Olympus cell Vivo IX 83), and further analyzed with the use of ImageJ^®^ software (v. 1.51j8, National Institutes of Health, Bethesda, MD, USA).

### 4.8. Immunoblotting

The cells were treated with 1,25(OH)_2_D_3_ as described above. After treatment, cells were collected; washed in PBS; resuspended in a RIPA lysis buffer (Sigma Aldrich) supplemented with protease and phosphatase inhibitor cocktail; and incubated for 30 min on ice with gentle shaking. The cell lysate was cleared by the centrifugation at 16,000× *g* for 10 min. Protein concentrations were determined by the Bradford assay. An equal number of lysates (40 µg) were loaded onto a 4–20% Mini-PROTEAN TGX Stain Free Gels (Bio-Rad, Hercules, CA, USA), resolved by SDS-PAGE, and transferred to a PVDF membrane using the Trans-Blot Turbo system (Bio-Rad). The membranes were incubated with primary antibodies anti-VDR (Santa Cruz Biotechnology, Dallas, TX, USA, sc-13133), anti-Cyp24 (Santa Cruz Biotechnology, sc-365700), anti-IL-33 (Bioss, bs-2208R) or anti-ST2 (Sigma-Aldrich, PRS3363) overnight at 4 °C. After washing, the membranes were incubated for 1 h at RT with an appropriate secondary peroxidase conjugated antibodies (anti-rabbit antibody, Santa Cruz Biotechnology, sc-2537 or anti-Mouse antibody, Sigma-Aldrich, A9044). Immunoreactive bands were developed using enhanced chemiluminescence ECL Plus (Perkin Elmer, Waltham, MA, USA, Cat. #NEL 103001EA). The membranes were stripped and reprobed using HRP conjugated anti-β-actin antibody (Santa Cruz Biotechnology, sc-47778) as a loading control. Changes in protein levels were assessed by densitometry of immunoreactive bands and followed by normalization relative to β-actin. 

### 4.9. Prediction of Transcription Factor Binding Sites

To predict potential transcription factor binding sites within IL33 and ST2 sequence upstream of the ATG start codon, we used the CiiiDER program from the Hudson Institute of Medical Research [53]. All the sequences were downloaded from the Ensembl website (https://www.ensembl.org/index.html, date of access: 27 July 2021) and scanned for transcription factor binding sites using JASPAR (https://jaspar.genereg.net, date of access: 27 July 2021) position frequency matrices (all binding sides in the database). In the next step, all predicted binding sites for VDR and its co-receptor RXR were analyzed, taking into account the matrix match score and positions in the sequence analysed. All data were visualized with the used of CiiiDER program. The predicted data were confirmed by in silico analysis of available ChiP-Seq data (H. sapiens (hg38), downloaded from ChiP-Atlas https://chip-atlas.org/peak_browser, date of access: 27 July 2021) and visualized by IGV browser [73].

### 4.10. Statistical Analysis

Data are presented as mean ± SD, and were analyzed with a Student’s *t*-test (for two groups) or one-way analysis of variance with appropriate post-hoc tests (for more than two groups). The data for patients are presented as the median value with range (min–max). Statistical analysis of immunostaining of skin samples based on measurements of fluorescence intensities was conducted by ImageJ^®^. From 5 to 7 random fields covering the epidermis for each specimen were analyzed using micrographs taken under 200× magnification. All statistical analyses were performed using GraphPad Prism v. 7.00 (GraphPad Software, San Diego, CA, USA). Statistically significant differences are denoted with asterisks: * *p* < 0.05, ** *p* < 0.01, *** *p* < 0.005.

## Figures and Tables

**Figure 1 ijms-22-12907-f001:**
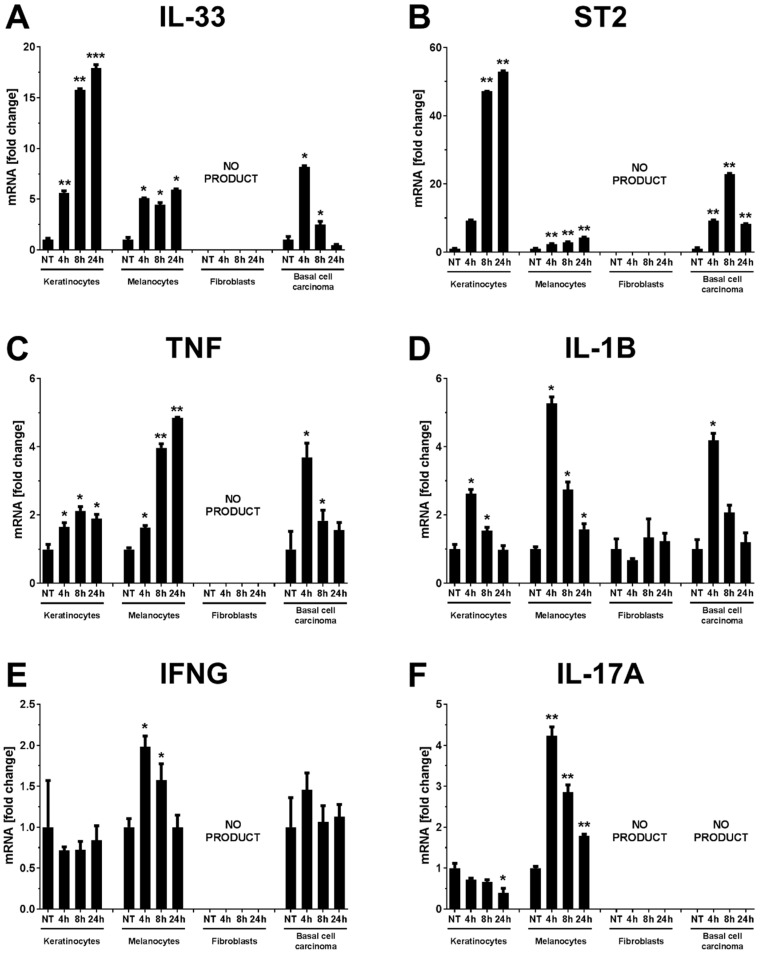
Vitamin D modulates the expression of *IL-33*, *ST2*, *TNF*, *IL-1B*, *IFNG* and *IL-17A* in skin cell lines. Keratinocytes (HPEKp), melanocytes (HEMas), fibroblasts (HDF) and basal cell carcinoma cells (A431) were stimulated with 1,25(OH)2D3 (100 nM) for 4, 8 and 24 h. The relative *IL-33* (**A**), *ST2* (**B**), *TNF* (**C**), *IL-1B* (**D**), *IFNG*, (**E**) and *IL-17A* (**F**) mRNA levels were analyzed by real-time PCR. * *p* < 0.05, ** *p* < 0.01, *** *p* < 0.001.

**Figure 2 ijms-22-12907-f002:**
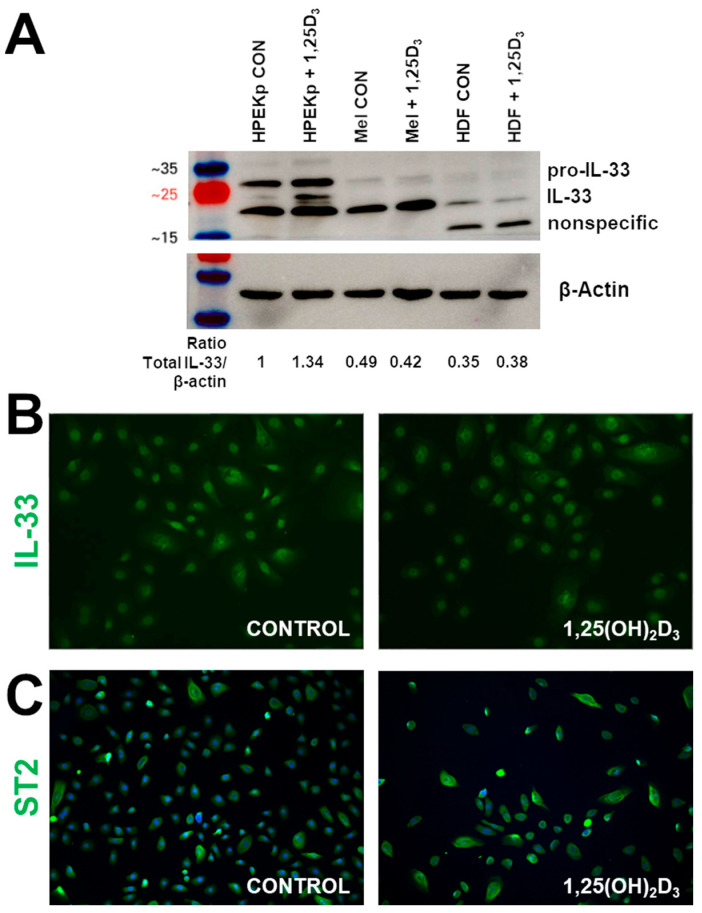
Vitamin D modulates the IL-33 and ST2 protein levels in keratinocytes. Keratinocytes were exposed to 100nM 1,25(OH)2D3 for 24 h. (**A**) Western blot of IL-33. The cumulative intensity of each band was quantified by densitometry analysis. The ratio of total IL-33 (pro-IL-33 and IL-33)/β-actin was shown below the blots. (**B**,**C**) Representative images of IF of HPEKp cells. A total of twenty-four hours after vitamin D stimulation, HPEKp were stained with antibodies against IL-33 (**B**) and ST2 (**C**). Magnification—200×.

**Figure 3 ijms-22-12907-f003:**
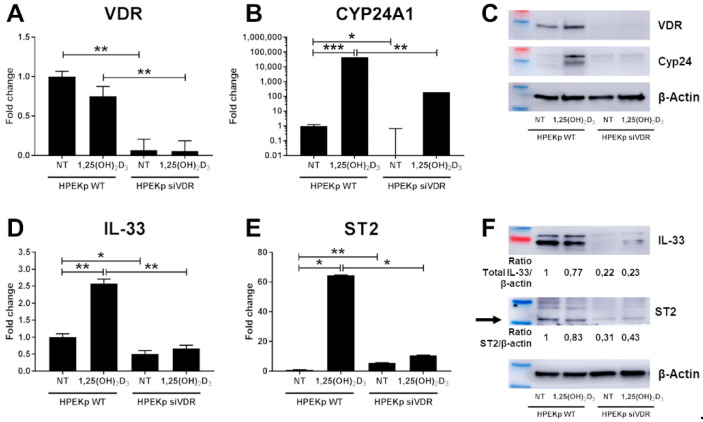
Knockdown of VDR in the human keratinocytes cell line (HPEKp) suppressed IL-33 and ST2 induction by 1,25(OH)_2_D_3_. (**A**,**B**) *VDR* (**A**), *CYP24A1* (**B**), expression 72 h after siRNA tranfecton was measured by real-time PCR in HPEKp cell line. After positive siRNA transfection, *IL-33* (**D**) and *ST2* (**E**) mRNA expression was analysed with or without 1,25(OH)_2_D_3_ (100 nM) stimulation. Immunoblotting for VDR, CYP24 (**C**), IL-33 and ST2 (**F**) was performed on lysate from HPEKp WT and siVDR cells with or without 1,25(OH)_2_D_3_ treatment for 24 h. Blots were stripped and re-probed with anti-β-actin antibody to normalize for differences in protein loading. * *p* < 0.05, ** *p* < 0.01, *** *p* < 0.001.

**Figure 4 ijms-22-12907-f004:**
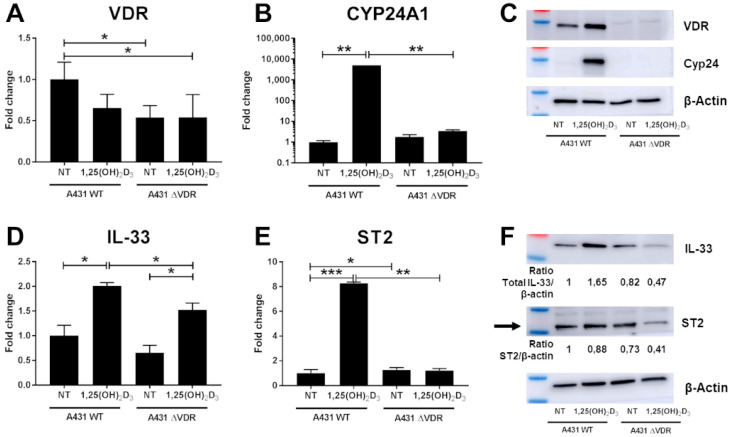
Knockout of VDR in the human basal cell carcinoma cell line (A431) suppressed IL-33 and ST2 induction by 1,25(OH)_2_D_3_. (**A**,**B**) The knockout of *VDR* (**A**), decreased expression *CYP24A1* (**B**), was confirmed by real-time PCR in A431 cell line. *IL-33* (**D**) and *ST2* (**E**) mRNA expression was examined in cells with or without 1,25(OH)_2_D_3_ (100 nM) treatment. Immunoblotting for VDR, CYP24 (**C**), IL-33 and ST2 (**F**) was performed on lysate from A431 WT and ΔVDR cells treated with or without 1,25(OH)_2_D_3_ for 24 h. Blots were stripped and re-probed with anti-β-actin antibody to normalize for differences in protein loading. * *p* < 0.05, ** *p* < 0.01, *** *p* < 0.001.

**Figure 5 ijms-22-12907-f005:**
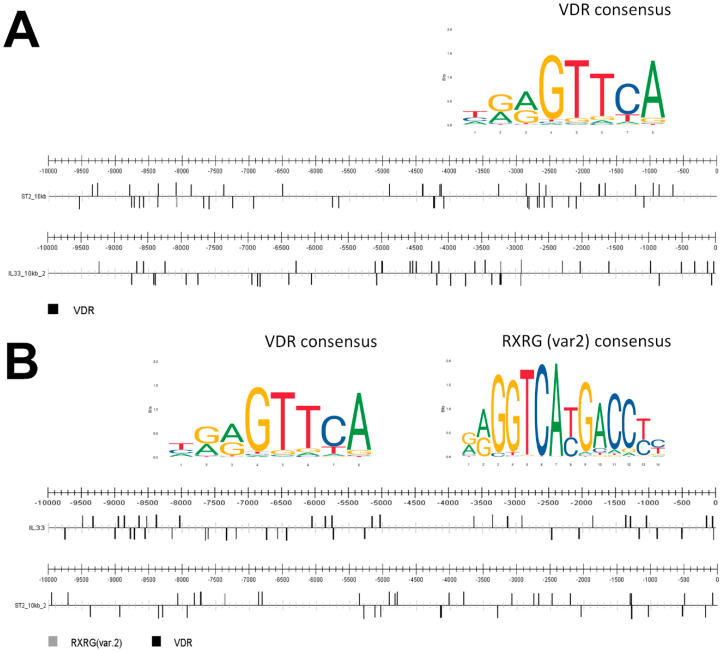
Prediction of VDR binding sites in IL-33 and ST2 genes by the CiiiDER program. (**A**) VDR binding sites (MA0693.2 JASPAR matrix) in the *IL-33* sequence 10 kb upstream of ATG, and a second 10 kb (10–20 kb) upstream of ATG. (**B**) VDR (MA0693.2 JASPAR matrix) and RXRGvar2 (MA1556.1 JASPAR matrix) binding sites in the *ST2* sequence 10 kb upstream of ATG, and a second 10 kb (10–20 kb) upstream of ATG.

**Figure 6 ijms-22-12907-f006:**
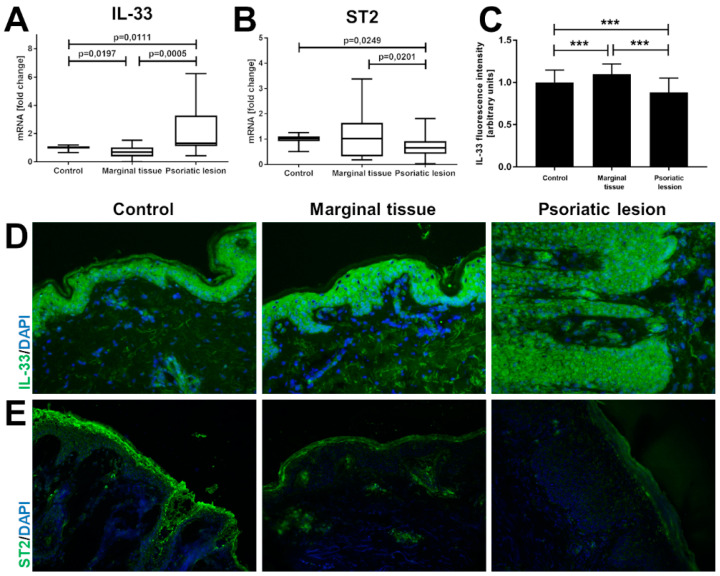
Changes in the expression of IL-33 and ST2 in psoriatic skin compared to controls at the mRNA and protein levels. Relative mRNA levels of IL-33 (**A**) and ST2 (**B**) were analyzed by quantitative real-time PCR in perilesional (*n* = 18) and lesional psoriatic skin (*n* = 18) compared with healthy control subjects (*n* = 13), as described in the methodology. Results are presented as boxes with min to max whiskers. Representative images of IF of psoriatic skin and healthy skin at 200× magnification are shown in (**D**,**E**). Skin samples were stained with rabbit polyclonal antibody to IL-33 (**C**,**D**) and rabbit polyclonal antibody to ST2 (**E**). *** *p* < 0.001.

**Table 1 ijms-22-12907-t001:** The list of PCR primers used in the study.

Gene Name	Forward Primer	Reverse Primer
*RPL37*	TTCTGATGGCGGACTTTACC	CACTTGCTCTTTCTGTGGCA
*IL-33*	GTGGAAGAACACAGCAAGCA	GTGGAAGAACACAGCAAGCA
*ST2*	CAGCACCTCTTGAGTGGTTTA	TCGCCGTCACACTATAATTGG
*TNF*	CCAGGGACCTCTCTCTAATCA	TCAGCTTGAGGGTTTGCTAC
*IL-1B*	ATCTCCGACCACCACTACA	GAAGGTGCTCAGGTCATTCTC
*IFNG*	GGAAAGAGGAGAGTGACAGAAA	TCATGTCTTCCTTGATGGTCTC
*IL-17A*	GGTTAAATGGACCCTGGCCT	ATAGTTGGTGTGCGGCTGAA
*VDR*	CTGTGGCAACCAAGACTACA	CATGCAAGTTCAGCTTCTTCAG
*CYP24A1*	GCAGCCTAGTGCAGATTT	ATTCACCCAGAACTGTTG

## Data Availability

The data that support the findings of this study are available from the corresponding author upon reasonable request.

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
