# Peer review of "The Effects of Vitamin D on the Expression of IL-33 and Its Receptor ST2 in Skin Cells; Potential Implication for Psoriasis"

_ijms, 2021, doi:10.3390/ijms222312907_

Round 1

Reviewer 1 Report

In this article, Justyna M. Wierzbicka, et al., provided the evidence to demonstrate the active form of vitamin D (1,25(OH)2D3) activates the mRNA expression of IL-33 and IL-33 receptor ST-2 in primary keratinocytes. Vitamin D also modulates the IL-33 protein level in keratinocytes. Vitamin D receptor VDR mediates the response of vitamin D-induced IL-33 and ST-2 expression.  They also provide the clinical data to demonstrate IL-33 and ST-2 expression pattern are significantly different in psoriatic lesion and healthy control in mRNA level and protein level. They concluded that vitamin D can modulate IL-33 signaling and open new perspectives for understanding of mechanisms of vitamin D action in psoriasis therapy. Although the paper is interesting, study strategy lacks reasonable rational and some results are confusing especially the results about IL-33 and ST-2 expression in psoriasis lesion and healthy control. Although the authors provide the additional data of vitamin D-treated basal cell carcinoma cell line A431 to demonstrate the effect of vitamin D on ST-2 expression is mediated by VDR and provide the putative VDR binding sites in IL-33 and ST-2 genes by program prediction in silico in this version, the authors still cannot provide a clear picture of their hypothesis. Also, the authors did not improve the data according to reviewer’s comment and current data cannot support the author’s final conclusion.

  1. In Figure 1, the authors demonstrate the different expression pattern of IL-33, ST-2, TNF alpha, IL-1B, INF-gamma and IL-17A in keratinocytes, melanocytes and fibroblasts with 1,25(OH)2D3 stimulation. What’s the biological or physiological role of IL-33/ST2 and analyzed cytokines in the 1,25(OH)2D3-treated skin cells especially keratinocytes and melanocytes. Does the expression of TNF alpha, IL-1B, INF-gamma and IL-17A regulate the mRNA expression of IL-33 and ST-2 in 1,25(OH)2D3-treated keratinocytes and melanocytes? In addition to examine the mRNA level of IL-33, TNF alpha, IL-1B, INF-gamma and IL-17A, it is necessary to analyze the protein level by ELISA and evaluate the protein level of ST-2 by immunoblotting.
  2. In Figure 2, the author examined the IL-33 protein level in keratinocytes with 1,25(OH)2D3. The protein amount of IL-33 is inconsistent with the mRNA level of IL-33 in three skin cell types. For example, 1,25(OH)2D3 induces IL-33 mRNA expression in melanocyte (shown in Figure 1), but does not change the IL-33 protein expression. Similarly, IL-33 mRNA cannot be detected in fibroblast (shown in Figure 1), but be detectible in protein level. The author needs to clarify the ~22kDa protein shown in 1,25(OH)2D3-treated keratinocytes indeed is cleaved IL-33. Additionally, why only the mature IL-33 could be detected in the melanocytes and fibroblasts? In the figure 2B, the intensity of immunofluorescence staining of IL-33 looks like very similar, this also inconsistent with the mRNA expression level and protein expression shown in Figure 2A. Finally, the immunofluorescence staining of ST-2 should be included.
  3. In Figure 3, the author only provided the mRNA level of VDR, CYP24A1, IL-33 and ST-2 in control and VDR knockdown keratinocyte. The protein level examined by immunoblotting is essential. In the Figure 3B, the SD bar is missing in the 1,25(OH)2D3-treated control and VDR knockdown keratinocyte. In addition, the author needs to describe the possible reason of ST2 expression increasing after VDR KD in keratinocytes and VDR KO in A431. Also, the figure legends of 3E and 3F are missing.
  4. It is hard to connect the results that 1,25(OH)2D3 activates the IL-33 and ST-2 expression in primary keratinocytes and melanocytes with the results of IL-33 upregulation and ST-2 downregulation in psoriatic lesion and get the conclusion that vitamin D can modulate IL-33 signaling and play a role in psoriasis therapy. It is helpful to use psoriasis animal model, such as imiquimod or IL-23 induced psoriasis like mice model, to examine the effect of 1,25(OH)2D3 on IL-33 expression during psoriasis progression may provide more informative data.
  5. In Figure 4, it is not enough to provide the prediction of VDR binding sites in IL-33 and ST-2 genes in silico. Experiments such as EMSA and reporter assay using wild type and mutant VDR to confirm the prediction are necessary.
  6. In Figure 5, the staining of ST-2 and its quantification should be included. The quantified IL-33 immunofluorescence intensity in Figure 5C and the presenting images in Figure 5D are inconsistent. In Figure 5D, what’s the difference between the upper panel and lower panel? The intensity of IL-33 of the upper panel and lower panel in Figure 5D is also inconsistent. More important, it looks like over-exposure images in Figure 5D and it is hard to distinguish the localization of IL-33 in cells as description in the result. The speculated infiltrating mononuclear cells should be clarified. Please add the scale in Figure 5D.

Author Response

Reviewer: 1

We thank the reviewer for evaluation of the manuscript and his/her insightful comments. According to the main suggestions we have added lacking stainings and Western Blots for ST2 protein

Comments and Suggestions for Authors

In this article, Justyna M. Wierzbicka, et al., provided the evidence to demonstrate the active form of vitamin D (1,25(OH)2D3) activates the mRNA expression of IL-33 and IL-33 receptor ST-2 in primary keratinocytes. Vitamin D also modulates the IL-33 protein level in keratinocytes. Vitamin D receptor VDR mediates the response of vitamin D-induced IL-33 and ST-2 expression.  They also provide the clinical data to demonstrate IL-33 and ST-2 expression pattern are significantly different in psoriatic lesion and healthy control in mRNA level and protein level. They concluded that vitamin D can modulate IL-33 signaling and open new perspectives for understanding of mechanisms of vitamin D action in psoriasis therapy. Although the paper is interesting, study strategy lacks reasonable rational and some results are confusing especially the results about IL-33 and ST-2 expression in psoriasis lesion and healthy control. Although the authors provide the additional data of vitamin D-treated basal cell carcinoma cell line A431 to demonstrate the effect of vitamin D on ST-2 expression is mediated by VDR and provide the putative VDR binding sites in IL-33 and ST-2 genes by program prediction in silico in this version, the authors still cannot provide a clear picture of their hypothesis. Also, the authors did not improve the data according to reviewer’s comment and current data cannot support the author’s final conclusion.

  • In Figure 1, the authors demonstrate the different expression pattern of IL-33, ST-2, TNF alpha, IL-1B, INF-gamma and IL-17A in keratinocytes, melanocytes and fibroblasts with 1,25(OH)2D3 stimulation. What’s the biological or physiological role of IL-33/ST2 and analyzed cytokines in the 1,25(OH)2D3-treated skin cells especially keratinocytes and melanocytes. Does the expression of TNF alpha, IL-1B, INF-gamma and IL-17A regulate the mRNA expression of IL-33 and ST-2 in 1,25(OH)2D3-treated keratinocytes and melanocytes? In addition to examine the mRNA level of IL-33, TNF alpha, IL-1B, INF-gamma and IL-17A, it is necessary to analyze the protein level by ELISA and evaluate the protein level of ST-2 by immunoblotting.

Thank you very much for that comment. Cytokines such as TNF, IL-1B, IFNG and IL-17A stimulate the production, or are the target of IL-33 as well as are important markers of psoriasis. It was not directly stated in the manuscript, therefore we have added those missing information with appropriate citations. Unfortunately, we were not able to perform ELISA experiments due to the limited time and lack of melanocytes and fibroblasts. Those cells and ELISA kits have to be purchase and we are not able in the given time for resubmission.

  • In Figure 2, the author examined the IL-33 protein level in keratinocytes with 1,25(OH)2D3. The protein amount of IL-33 is inconsistent with the mRNA level of IL-33 in three skin cell types. For example, 1,25(OH)2D3 induces IL-33 mRNA expression in melanocyte (shown in Figure 1), but does not change the IL-33 protein expression. Similarly, IL-33 mRNA cannot be detected in fibroblast (shown in Figure 1), but be detectible in protein level. The author needs to clarify the ~22kDa protein shown in 1,25(OH)2D3-treated keratinocytes indeed is cleaved IL-33. Additionally, why only the mature IL-33 could be detected in the melanocytes and fibroblasts? In the figure 2B, the intensity of immunofluorescence staining of IL-33 looks like very similar, this also inconsistent with the mRNA expression level and protein expression shown in Figure 2A. Finally, the immunofluorescence staining of ST-2 should be included.

Thank you very much for that comment.

We wish it could be a simple explanation and very elegant correlation between mRNA and protein levels, but biological systems are more complex and processes of transcription and translation in eukaryotes are regulated separately. Furthermore, both mRNA and proteins undergo degradation. Thus, it is possible that upregulation of mRNA level may not results in more protein. For example, a fast degradation of given protein may trigger its mRNA synthesis. In order to solve this issue, we would have to run separate project. Thus, all speculations concerning mechanisms were removed from the paper. The novelty of this project is demonstration that vitamin D can regulate expression of IL-33 and its receptor and the effect is vitamin D receptor dependent. Furthermore, it seems that keratinocytes produce IL-33 and its receptors, but this expression is modulated in psoriasis suggestion involvement of keratinocytes and IL-33 triggered pathways in its pathogenesis. However, we are fully aware, that more study is needed to describe the mechanisms underlying these interactions. Please, see my comments above concerning mRNA vs protein levels.

Response concerning multiple bends for IL-33 on WB.

IL-33 is synthesized as a 30-kDa precursor and subsequently activated proteolitically by caspases (Biol Chem. 2009 Jul 17;284(29):19420-6). Thus, it could be assumed that the higher band represent precursor form, lower band(s) activated form(s). However, many factors may influence proteolytic modification of IL-33, thus even several variants of IL-33 protein with molecular weight varied form 18-21 kDa could be expected (PNAS January 31, 2012 109 (5) 1673-1678; PNAS June 2, 2009 106 (22) 9021-9026). Finally, proteolytic of IL-33 is not a main scope of this project, thus it was not investigated nor discussed.

The immunofluorescence staining of ST-2 for HPEKp cell line was included.

  • In Figure 3, the author only provided the mRNA level of VDR, CYP24A1, IL-33 and ST-2 in control and VDR knockdown keratinocyte. The protein level examined by immunoblotting is essential. In the Figure 3B, the SD bar is missing in the 1,25(OH)2D3-treated control and VDR knockdown keratinocyte. In addition, the author needs to describe the possible reason of ST2 expression increasing after VDR KD in keratinocytes and VDR KO in A431. Also, the figure legends of 3E and 3F are missing.

Thank you very much for that comment. We have added all missing Western Blots for both HPEKp and A431 cell lines. We have also divided this Figure 3 into two smaller ones – Figure 3 is showing all the results for HPEKp cell line, while Figure 4 for A431.

SD bar is not missing in the CYP24A1 graphs, it is just relatively small comparing to the fold change of 1,25(OH)2D3-treated samples. Because of that we are using logarithmic y-axis scale.

  • It is hard to connect the results that 1,25(OH)2D3 activates the IL-33 and ST-2 expression in primary keratinocytes and melanocytes with the results of IL-33 upregulation and ST-2 downregulation in psoriatic lesion and get the conclusion that vitamin D can modulate IL-33 signaling and play a role in psoriasis therapy. It is helpful to use psoriasis animal model, such as imiquimod or IL-23 induced psoriasis like mice model, to examine the effect of 1,25(OH)2D3 on IL-33 expression during psoriasis progression may provide more informative data.

Thank you very much for that comment. Suggested animal experiments represent straightforward continuation of our study, however this would be a topic of separated project which requires separate financial support.

  • In Figure 4, it is not enough to provide the prediction of VDR binding sites in IL-33 and ST-2 genes in silico. Experiments such as EMSA and reporter assay using wild type and mutant VDR to confirm the prediction are necessary.

Thank you very much for that comment. Please, note that prediction of VDR binding sites in IL-33 and ST-2 are just adds-on to the manuscript and due to lack of direct confirmation are treated as hypothetical, however we are fully aware that final prove is still needed, but we believe that EMSA, reporter assays as well as sided directed mutagenesis of promotors could be a subject of separate study.

  • In Figure 5, the staining of ST-2 and its quantification should be included. The quantified IL-33 immunofluorescence intensity in Figure 5C and the presenting images in Figure 5D are inconsistent. In Figure 5D, what’s the difference between the upper panel and lower panel? The intensity of IL-33 of the upper panel and lower panel in Figure 5D is also inconsistent. More important, it looks like over-exposure images in Figure 5D and it is hard to distinguish the localization of IL-33 in cells as description in the result. The speculated infiltrating mononuclear cells should be clarified. Please add the scale in Figure 5D.

Thank you very much for that comment. We have added missing ST2 staining of the control and psoriatic skin, it replaced previous 5E panel (higher magnification of IL-33 staining). Now both panels (5D and 5E) are showing 2 different staining’s in the same magnification (200x). Unfortunately, we were not able to add a scale bars to the pictures because those photos were taken by different fluorescent microscopes, therefore we added appropriate information in the Figure legend. Finally, speculation concerning mononuclear cells and their localisation has been clarified as follow:

“It seems that mononuclear cells infiltrating perivascular region of dermis,” (Page 18, Line 3) – those cells could be clearly seen in reticular part of the dermis next to blood vessels and those cells are also show immunoreactivity characteristic to IL-33 (see figure 6).

Reviewer 2 Report

I read with great interest this manuscript pointing out the effect of vitamin D in the expression of IL-33, a cytokine belonging to the IL-1 family.

I think that it is a valuable article but needs some minor corrections:

In the first sentence of the introduction please add this references that sustain the statement:

"Beyond the effects on bone homeostasis, vitamin D exerts impact on a variety of other organs or systems, including the skin as well as immune system. Keratinocytes are not only a natural source of vitamin D, but due to the expression of vitamin D receptor (VDR), they are also target cells which respond in an autocrine and paracrine manner to its bio-logically active form.

Please better characterize psoriasis comorbidities adding cardiovascular, respiratory and gastrointestinal ones relative info. with reference since they exert a great effect on skin flares.

Please in the discussion highlight the potential magnitude of your discovery in the realize management of pruritus in psoriasis  [10.1111/jdv.15539, 10.1080/09546634.2020.1840502] and its influence on circadian rhythm [10.1080/07420528.2019.1678629].

Author Response

We thank the reviewer for evaluation of the manuscript and his/her insightful comments.

According to the suggestions of the Reviewer, we added all missing citations and tried to better describe psoriasis comorbidities.

Comments and Suggestions for Authors

I read with great interest this manuscript pointing out the effect of vitamin D in the expression of IL-33, a cytokine belonging to the IL-1 family.

I think that it is a valuable article but needs some minor corrections:

In the first sentence of the introduction please add this references that sustain the statement:

"Beyond the effects on bone homeostasis, vitamin D exerts impact on a variety of other organs or systems, including the skin as well as immune system. Keratinocytes are not only a natural source of vitamin D, but due to the expression of vitamin D receptor (VDR), they are also target cells which respond in an autocrine and paracrine manner to its bio-logically active form.

Please better characterize psoriasis comorbidities adding cardiovascular, respiratory and gastrointestinal ones relative info. with reference since they exert a great effect on skin flares.

Please in the discussion highlight the potential magnitude of your discovery in the realize management of pruritus in psoriasis  [10.1111/jdv.15539, 10.1080/09546634.2020.1840502] and its influence on circadian rhythm [10.1080/07420528.2019.1678629].

Round 2

Reviewer 1 Report

The author well answer the questions and improve their data.

Author Response

Thank you very much for your time and report. We have made correction on English language.

This manuscript is a resubmission of an earlier submission. The following is a list of the peer review reports and author responses from that submission.

Round 1

Reviewer 1 Report

In this report, the authors focus on the gene and protein expressions of IL-33 and ST2 in cultured skin cells induced by an active form of vitamin D, 1,25(OH)2D3 (calcitriol), in skin tissues from psoriatic patients. The authors suggest that the modulation of IL-33 signaling is a new mechanism of vitamin D in psoriasis therapy. Indeed, the effects of 1,25(OH)2D3 on the gene expression of IL-33 and ST2 in keratinocytes may be interesting. However, whether the effects of vitamin D on IL-33-ST2 axis are involved in its clinical efficacy for psoriasis is not fully addressed, and therefore it is difficult to accept their suggestion. Furthermore, the role of IL-33-ST2 axis is also still unclear, although the authors propose the dysregulation of the expression of IL-33 and ST2 greatly contribute to the pathogenesis of psoriasis.

Major comments

  1. Page 2, line 69. The statement “atopic dermatitis, where it is undergoing alternative splicing [30-33], as well as psoriasis” is incorrect. Alternative splicing of IL-33 in atopic dermatitis has not been demonstrated.
  2. Fig. 1: To compare the expression levels of cytokines and ST2 among different cell types, the authors should present ELISA data for them, because real-time PCR cannot reveal total amount of transcripts for those genes. Data for melanocytes and fibroblasts may be unnecessary.

  3. Fig. 1: 1,25(OH)2D3 induces the expression of IL-33 gene more than 50-fold in cultured keratinocytes HPEKp and lesser extent in cultured melanocytes HEMas at 24 hours of incubation. However, western blotting showed only 34% induction of IL-33 in HPEKp with unusual extra-band. The authors should clarify the discrepancy between those results.

  4. Fig. 3: If the 1,25(OH)2D3-induced IL-33 is not released, what is the role of IL33? The authors suggested that calcitriol may function by maintaining keratinocytes in a steady-state condition. If so, provide evidence that IL-33 regulates any cellular functions of keratinocytes in result section.
  5. It is well known that soluble ST2 is up-regulated by 1,25(OH)2D3 (Pfeffer PE, Chen YH, Woszczek G, et al. Vitamin D enhances production of soluble ST2, inhibiting the action of IL-33. J Allergy Clin Immunol. 2015;135(3):824-7.e3), although the report is not cited in this manuscript. ST2 has two splice variants encoding membrane-bound and soluble form. The expression of both transcripts should be examined for presenting the gene expression of ST2. Which splice variant was induced in keratinocytes by 1,25(OH)2D3?

  6. Fig. 4: Immunofluorescence of control and psoriatic skins using an IL-33 antibody was not clear with high background, and the differences among samples seem to less than 20%, which may be within error margins. For proper comparisons, the authors should use confocal imaging to record data.

  7. The expression of IL-33 gene was elevated in psoriatic lesions, whereas ST2 was down-regulated (Fig. 4). According to the results, the authors state “Our results indicate that there is significant dysregulation among the expression of both, IL-33 and ST2, in psoriasis which may greatly contribute to its pathogenesis” in discussion section on page 8, line 247. If so, the authors should provide any evidence that this dysregulation is involved in psoriasis.

  8. In discussion section on page 8 line 244, the authors also state “it is possible that vitamin D may restore the balance between IL-33 and ST2 which is necessary for the maintenance of proper psoriatic immune response”. However, to suggest that, the authors should provide additional evidence, for example, using animal models that psoriatic lesions induced by imiquimod or IL-23 are ameliorated or exacerbated in IL-33 or ST2 knockout mice.

  9. What mononuclear cells express IL-33 (on Page 8, line 228)? The authors should characterize those cells at least using immunohistochemistry.

  10. It may be interesting that vitamin D induces the gene expression of IL-33 and ST2 in keratinocytes, but its role as a therapeutic agent for psoriasis has not fully addressed been in the present study. Therefore, it is difficult to conclude that the effect of vitamin D on IL-33-ST2 axis is involved in the therapeutic mechanism of vitamin D for psoriasis, even if the increase and decrease of IL-33 and ST2 in psoriatic or marginal lesions.

Minor comments

  1. In text, Figure legend and Figures: Gene names and protein names should be correctly described according to the official nomenclature.

  2. 1,25(OH)2D3 is an abbreviation of “1,25-Dihydroxyvitamin D3”. Its drug name is “calcitriol”. Why both terms are used randomly in the manuscript? It is confusing for readers.

  3. Fig. 2: Lane captions are mismatched.

  4. Provide the incubation time for 1,25(OH)2D3 in Fig. 3 legend.

  5. Fig. 4D: What do arrows and asterisks mean? What are differences of samples in upper and lower panels?

  6. Page 10, line 325. Is Nikon Eclipse E800 a fluorescence microscope?

  7. Mistyping: Page 8, line 229:“aggregation” may be “aggravation”; Page 8, line 233:“disease free” should be “disease-free”; Page 8, line 241: “psoriatic plague” may be “psoriatic plaque”.

Reviewer 2 Report

In this article, Justyna M. Wierzbicka, et al., provided the evidence to demonstrate the active form of vitamin D (1,25(OH)2D3) activates the mRNA expression of IL-33 and IL-33 receptor ST-2 in primary keratinocytes. Vitamin D also modulates the IL-33 protein level in keratinocytes. Vitamin D receptor VDR mediates the response of vitamin D-induced IL-33 and ST-2 expression.  They also provide the clinical data to demonstrate IL-33 and ST-2 expression pattern are significantly different in psoriatic lesion and healthy control in mRNA level and protein level. They concluded that vitamin D can modulate IL-33 signaling and open new perspectives for understanding of mechanisms of vitamin D action in psoriasis therapy. Although the paper is interesting, study strategy lacks reasonable rational and some results are confusing especially the results about IL-33 and ST-2 expression in psoriasis lesion and healthy control. The author cannot provide a clear picture of their hypothesis. Also, current data cannot support the author’s final conclusion.

  1. In Figure 1, the author demonstrate the different expression pattern of IL-33, ST-2, TNF alpha, IL-1B, INF-gamma and IL-17A in keratinocytes, melanocytes and fibroblasts with 1,25(OH)2D3 stimulation. What’s the biological or physiological role of IL-33/ST2 and analyzed cytokines in the 1,25(OH)2D3-treated skin cells especially keratinocytes and melanocytes. Does the expression of TNF alpha, IL-1B, INF-gamma and IL-17A regulate the mRNA expression of IL-33 and ST-2 in 1,25(OH)2D3-treated keratinocytes and melanocytes? In addition to examine the mRNA level of IL-33, TNF alpha, IL-1B, INF-gamma and IL-17A, it is necessary to analyze the protein level by ELISA and evaluate the protein level of ST-2 by immunoblotting or cell-surface staining and quantified by flow cytometry.
  2. In Figure 2, the author examined the IL-33 protein level in keratinocytes with 1,25(OH)2D3. The protein amount of IL-33 is inconsistent with the mRNA level of IL-33 in three skin cell types. For example, 1,25(OH)2D3 induces IL-33 mRNA expression in melanocyte (shown in Figure 1), but does not change the IL-33 protein expression. Similarly, IL-33 mRNA cannot be detect in fibroblast (shown in Figure 1), but be detectible in protein level. The author need to clarify the 25kDa protein shown in 1,25(OH)2D3-treated keratinocyte indeed is cleaved IL-33. Additionally, why only the mature IL-33 could be detected in the melanocytes and firoblasts? In the figure 2B, the intensity of immunofluorescence staining of IL-33 looks like very similar, this also inconcictent with the mRNA expression level and protein expression shown in Figure 2A. Finally, the immunofluorescence staining of ST-2 should be included. Please align each treatment markers with each sample loading line of blotting membrane in Figure 2A and add the scale in Figure 2B.
  3. In Figure 3, the author only provided the mRNA level of VDR, CYP24A1, IL-33 and ST-2 in control and VDR knockdown keratinocyte. The protein level examined by immunoblotting is essential. In the Figure 3B, the SD bar is missing in the 1,25(OH)2D3-treated control and VDR knockdown keratinocyte.
  4. It is hard to connect the results that 1,25(OH)2D3 activates the IL-33 and ST-2 expression in primary keratinocytes and melanocytes with the results of IL-33 upregulation and ST-2 downregulation in psoriatic lesion and get the conclusion that vitamin D can modulate IL-33 signaling and paly a role in psoriasis therapy. It is helpful to use psoriasis animal model, such as imiquimod or IL-23 induced psoriasis like mice model, to examine the effect of 1,25(OH)2D3 on IL-33 expression during psoriasis progression may provide more informative data.
  5. In Figure 4, the staining of ST-2 and its quantification should be included. The quantified IL-33 immunofluorescence intensity in Figure 4C and the presenting images in Figure 4D are in consistent. In Figure 4D, what’s the difference between the upper panel and lower panel? What do the arrow and star symbol indicate? More important, it looks like over-exposure images in Figure 4D and it is hard to distinguish the localization of IL-33 in cells as description in the result. The speculated infiltrating mononuclear cells should be clarified. Please add the scale in Figure 4D.